# Hotspot DNA Methyltransferase 3A (*DNMT3A*) and Isocitrate Dehydrogenase 1 and 2 (*IDH1/2*) Mutations in Acute Myeloid Leukemia and Their Relevance as Targets for Immunotherapy

**DOI:** 10.3390/biomedicines12051086

**Published:** 2024-05-14

**Authors:** Nadine E. Struckman, Rob C. M. de Jong, M. Willy Honders, Sophie-Anne I. Smith, Dyantha I. van der Lee, Georgia Koutsoumpli, Arnoud H. de Ru, Jan-Henrik Mikesch, Peter A. van Veelen, J. H. Frederik Falkenburg, Marieke Griffioen

**Affiliations:** 1Department of Hematology, Leiden University Medical Center, 2333ZA Leiden, The Netherlands; n.e.struckman@lumc.nl (N.E.S.); rcmdejong@gmail.com (R.C.M.d.J.); m.w.honders@lumc.nl (M.W.H.); s.i.smith@lumc.nl (S.-A.I.S.); d.i.van_der_lee@lumc.nl (D.I.v.d.L.); g.koutsoumpli@lumc.nl (G.K.); j.h.f.falkenburg@lumc.nl (J.H.F.F.); 2Center for Proteomics and Metabolomics, Leiden University Medical Center, 2333ZA Leiden, The Netherlands; a.h.de_ru@lumc.nl (A.H.d.R.); p.a.van_veelen@lumc.nl (P.A.v.V.); 3Department of Medicine A, University Hospital Münster, 48149 Münster, Germany; jan-henrik.mikesch@ukmuenster.de

**Keywords:** cancer immunotherapy, acute myeloid leukemia, neoantigen, DNA methyltransferase 3A, isocitrate dehydrogenase 2, hotspot mutation, T-cell therapy

## Abstract

DNA methyltransferase 3A (*DNMT3A*) and isocitrate dehydrogenase 1 and 2 (*IDH1/2*) are genes involved in epigenetic regulation, each mutated in 7–23% of patients with acute myeloid leukemia. Here, we investigated whether hotspot mutations in these genes encode neoantigens that can be targeted by immunotherapy. Five human B-lymphoblastoid cell lines expressing common HLA class I alleles were transduced with a minigene construct containing mutations that often occur in *DNMT3A* or *IDH1/2*. From these minigene-transduced cell lines, peptides were eluted from HLA class I alleles and analyzed using tandem mass spectrometry. The resulting data are available via ProteomeXchange under the identifier PXD050560. Mass spectrometry revealed an HLA-A*01:01-binding DNMT3A^R882H^ peptide and an HLA-B*07:02-binding IDH2^R140Q^ peptide as potential neoantigens. For these neopeptides, peptide–HLA tetramers were produced to search for specific T-cells in healthy individuals. Various T-cell clones were isolated showing specific reactivity against cell lines transduced with full-length *DNMT3A^R882H^* or *IDH2^R140Q^* genes, while cell lines transduced with wildtype genes were not recognized. One T-cell clone for DNMT3A^R882H^ also reacted against patient-derived acute myeloid leukemia cells with the mutation, while patient samples without the mutation were not recognized, thereby validating the surface presentation of a DNMT3A^R882H^ neoantigen that can potentially be targeted in acute myeloid leukemia via immunotherapy.

## 1. Introduction

Acute myeloid leukemia (AML) is the most common type of leukemia in adults. Despite the high rate of initial complete remission in AML patients treated with intensive chemotherapy, relapses frequently occur [1]. Allogeneic hematopoietic stem cell transplantation has been established as a treatment that effectively reduces the risk of relapse and increases the overall survival of AML patients [2]. The anti-tumor effect of allogeneic stem cell transplantation is induced by strong alloreactive T-cell responses, indicating that cellular immunotherapy has the potential to eradicate AML [3]. Although allogeneic stem cell transplantation increases overall survival, up to half of AML patients still relapse [4], emphasizing the need to develop new treatment strategies for AML.

We and others have shown that neoantigens encoded by recurrent genetic aberrations in AML can be targeted using immunotherapy [5,6,7,8,9,10]. Neoantigens are attractive targets for therapy since they are highly specific for cancer cells [11]. We previously isolated a T-cell receptor (TCR) for an HLA-A*02:01-binding neoantigen derived from mutant NPM1 [5]. T-cells engineered with this TCR have been shown to effectively target AML in vitro as well as in mice engrafted with AML. 

In AML, mutations have been found in genes involved in epigenetic regulation, including genes affecting DNA (hydroxy)methylation, such as DNA methyltransferase 3A (*DNMT3A*), Tet oncogene family member 2 (*TET2*), and isocitrate dehydrogenase 1 and 2 (*IDH1/2*). Mutations in *DNMT3A*, *TET2*, *IDH1*, and *IDH2* each occur in 7–23% of AML cases [12], and are also found in clonal hematopoiesis of indeterminate potential (CHIP), a condition prevalent in elderly people and patients in remission after treatment for AML [13]. Individuals with CHIP are at increased risk of developing AML. *DNMT3A*, *TET2*, *IDH1*, and *IDH2* mutations in hematopoietic stem cells from individuals with CHIP are usually sub-clonal. Targeting these mutations in patients with AML may not only effectively eradicate leukemia, but also pre-malignant hematopoietic stem cells, thereby reducing the risk of relapse.

Neoantigen identification benefits greatly from human cell line models where somatic mutations are introduced via Clustered Regularly Interspaced Short Palindromic Repeats (CRISPRs) and CRISPR-associated protein 9 (Cas9), thereby allowing analysis of neoantigen surface expression under more physiological conditions [14]. Conventional CRISPR techniques use complexes consisting of Cas9 and a custom-designed single-guide RNA (sgRNA), which guides Cas9 to the target locus, where it induces a double-stranded DNA break. These breaks are repaired through mechanisms like nonhomologous end joining or homology-directed repair, which facilitates the introduction of a repair template. Base editing is a new CRISPR method that enhances precision genome editing. It involves a Cas9 nickase, which creates single-stranded breaks, and an enzyme that chemically modifies a single base, which is corrected by endogenous cellular repair mechanisms. Unlike homology-directed repair, which is limited to dividing cells, base editing can modify both dividing and nondividing cells [15].

Here, we explored whether mutations in genes involved in DNA methylation or hydroxymethylation encode neoantigens that can be targeted with immunotherapy. We focused on hotspot mutations in *DNMT3A*, *IDH1*, and *IDH2* [12], as they may encode recurrent neoantigens that are shared between multiple patients. Two HLA class I-binding neopeptides derived from DNMT3A^R882H^ and IDH2^R140Q^ were identified using tandem mass spectrometry, and used to search for specific T-cell clones. Isolated T-cell clones were screened against human cell lines expressing a natural *DNMT3A^R882H^* mutation [16] or *IDH2^R140Q^* mutation introduced by CRISPR-Cas9 using homology-directed repair [17]. Among the various isolated T-cell clones, one clone showed specific reactivity against patient-derived AML cells with *DNMT3A^R882H^*, thereby validating the surface presentation of a neoantigen that can potentially be targeted with immunotherapy.

## 2. Materials and Methods

### 2.1. Human Samples and Cell Culture

Peripheral blood and bone marrow samples from AML patients were acquired from the Leiden University Medical Center Biobank for Hematological Diseases. Peripheral blood mononuclear cells (PBMCs) used for T-cell isolations were freshly used and obtained from healthy individuals (Sanquin, Amsterdam, The Netherlands). PBMCs and bone marrow mononuclear cells were isolated using Ficoll-Isopaque density gradient centrifugation. Patient AML samples were thawed and incubated in Iscove’s Modified Dulbecco’s Medium (IMDM) supplemented with 10% human serum (Sanquin) and 100 U/mL penicillin/streptomycin (pen/strep; Lonza, Basel, Switzerland). Epstein–Barr virus-transformed human B-lymphoblastoid cell lines (EBV-B cell lines), K562, and HL-60 cell lines (ATCC, Manassas, MA, USA) were cultured in IMDM supplemented with fetal bovine serum (FBS; Thermo Fisher Scientific, Bremen, Germany), 200mM L-glutamine (Lonza), and pen/strep. The human K562-R140Q cells had a homozygous *IDH2^R140Q^* mutation introduced via CRISPR-Cas9-guided genome editing [17]. SET-2 (DSMZ), which is an AML cell line with a heterozygous *DNMT3A^R882H^* mutation [16], was cultured in Dulbecco’s Modified Eagle Medium (DMEM; Lonza) supplemented with FBS, L-glutamine, pen/strep, and 1 × 10^5^ M β-mercaptoethanol (Sigma Aldrich, Burlington, MA, USA). T-cells were cultured in T-cell medium (TCM) consisting of IMDM with FBS, human serum, L-glutamine, pen/strep, and 100 IU/mL IL-2 (Novartis, Macquarie Park, NSW, Australia). T-cells were (re)stimulated every 14 days with 0.8 µg/mL phytohemagglutinin (PHA) and irradiated (40 Gy) allogeneic feeder cells.

### 2.2. Gene Constructs

*DNMT3A*, *IDH1*, and *IDH2* hotspot mutations were combined as minigenes in an MP71 retroviral construct containing the nerve growth factor receptor (NGFR) as a marker gene. Each minigene consisted of 87 nucleotides coding for 29-mer peptides with the mutated amino acid at position 15. The construct contained the following minigenes from the 5′ to 3′ ends: *DNMT3A c.2645G>A*, *IDH2 c.419G>A*, *IDH1 c.395G>A*, *IDH1 c.394C>T*, *DNMT3A c.2644C>T*, *IDH2 c.515G>A*, *IDH1 c.394C>G*, and *IDH1 c.394C>A*. Constructs with full-length *DNMT3A^R882H^* or *IDH2^R140Q^* genes were inserted into the LZRS retroviral vector containing NGFR as a marker gene. Genes encoding HLA-A*01:01 or HLA-B*07:02 were inserted into the LZRS vector containing mouse CD19 as a marker gene. After retroviral transduction, EBV-B cell lines, K562-R140Q, and SET-2 cells were purified on NGFR or mouse CD19 expression using magnetic beads (MACS beads; Miltenyi Biotec, Westphalia, Germany).

### 2.3. Samples for Peptide Elution

To identify the neopeptides encoded by *DNMT3A*, *IDH1*, or *IDH2* hotspot mutations, cell pellets (1 × 10^9^ cells) were collected from five EBV-B cell lines transduced with the minigene construct (Appendix A), as well as from SET-2 transduced with HLA-A*01:01 (and the full-length *DNMT3A^R882H^* gene) and K562-R140Q transduced with HLA-B*07:02 (and the full-length *IDH2^R140Q^* gene). In addition, cell pellets (100 × 10^6^ cells) from two patient AML samples were collected. AML samples were selected based on the HLA type and *DNMT3A* or *IDH2* mutations as identified via RNA-Seq and validated with Sanger sequencing [18].

### 2.4. HLA Class I Peptide Elution

Peptide elution was performed as described previously [5]. Briefly, cell pellets were lysed and HLA class I molecules were immunoprecipitated using an immunoaffinity column to extract peptide–HLA (pHLA) complexes using an HLA class I antibody (W6/32, ATCC). pHLA complexes were dissociated from the column using 10% acetic acid. Peptides and HLA class I molecules were separated via filtration through a 10 kDa membrane. The collected peptide pools were fractionated with strong cation exchange chromatography (SCX) and analyzed using mass spectrometry.

### 2.5. Tandem Mass Spectrometry

For data-dependent acquisition (DDA) tandem mass spectrometry (MS/MS), the peptides were lyophilized, dissolved in a solution of 95/3/0.1 *v*/*v*/*v* water/acetonitrile/formic acid, and analyzed online with C18 nano-HPLC MS/MS using an Easy nLC 1200 gradient HPLC system (Thermo, Bremen, Germany) and a LUMOS mass spectrometer (Thermo). Fractions were injected onto a homemade precolumn (100 μm × 15 mm; Reprosil-Pur C18-AQ 3 μm, Dr. Maisch, Ammerbuch, Germany) and eluted via a homemade analytical nano-HPLC column (30 cm × 50 μm; Reprosil-Pur C18-AQ 3 μm). The gradient ran from 2% to 36% solvent B (20/80/0.1 water/acetonitrile/formic acid (FA) *v*/*v/v*) over 120 min. The nano-HPLC column served as the electrospray needle of the MS source, with a tip diameter of ~5 μm. The LUMOS mass spectrometer operated in the data-dependent MS/MS mode with a cycle time of 3 s, HCD collision energy at 32 V, and recording of the MS2 spectrum in the orbitrap. In the master scan (MS1), the resolution was set at 60,000× and the scan range was from 300 to 1400, with a ‘standard’ AGC target and a maximum fill time of 50 ms. Dynamic exclusion after *n* = 1 with an exclusion duration of 20 s was applied. Charge states included 1 (precursor selection range: 800–1400), 2 (precursor selection range: 400–800), and 3 (precursor selection range: 300–600). For MS2, precursors were isolated with the quadrupole using an isolation width of 1.2 Da. The MS2 scan resolution was 30,000 with the ‘standard’ AGC target at a ‘dynamic’ maximum fill time. For the post-analysis processing, raw data were first converted to peak lists using Proteome Discoverer version 2.1 (Thermo Electron), and then submitted to the Uniprot Homo sapiens minimal database (20,596 entries), using Mascot v. 2.2.07 (www.matrixscience.com) for protein identification. Mascot searches were carried out with 10 ppm and 0.02 Da deviation for the precursor and fragment mass, respectively, and no enzyme was specified. Methionine oxidation and cysteinylation of cysteine were set as variable modifications. The false discovery rate was set to <1% and, in addition, peptides with mascot ion scores <35 were discarded. We searched for 9, 10, and 11mer peptides matching amino acid sequences encoded by *DNMT3A* or *IDH2* hotspot mutations, but not wildtype *DNMT3A* or *IDH2* sequences. NetMHCpan version 4.1 was used to predict peptide binding to common HLA alleles [19].

For parallel reaction-monitoring mass spectrometry (PRM-MS), the samples were lyophilized and resuspended in buffer A containing 4 fmol/uL of each heavy peptide. HLA–eluates were injected together with a mix of heavy labeled peptides (20 fmol each). The Orbitrap Fusion LUMOS mass spectrometer was operated in the PRM mode. DNMT3A (YTD**V**SNMSH, YTDVSNMSH**L**A) and IDH2 (S**P**NGTIQNIL) peptides were monitored (heavy amino acids in bold and underlined). Appendix A shows the selected peptides, transitions, and collision energies. The Q1 isolation width was 1.2 Da and the MS2 resolution was 30,000 at an AGC target value of 1 million and with a maximum fill time of 100 ms. The gradient was the same as that used for the discovery mode experiments. PRM data analysis and data integration were performed in Skyline 3.6.0.10493. 

### 2.6. Antibodies, Peptide–HLA Tetramers, and Flow Cytometry

For T-cell isolation, PBMCs were stained with CD8 Alexa Fluor 700 (Invitrogen (Waltham, MA, USA)/MHCD0829), CD4 FITC (BD (Franklin Lakes, NJ, USA)/555346), CD14 FITC (BD/555397), CD19 FITC (BD/555412), and in-house generated PE-conjugated pHLA tetramers produced as outlined previously with minor modifications [20]. T-cell clones were stained with pHLA tetramers for 15 min at room temperature. AML samples were stained with HLA class I FITC (Bio-Rad (Hercules, CA, USA)/MCA81), CD54/ICAM1 APC Fire 750 (BioLegend (San Diego, CA, USA)/353121), CD58/LFA-3 BUV395 (BD/752794), and CD102/ICAM2 BB700 (BD/746179). For the cytotoxicity assays, the AML cells were stained with CD33 PE (Biolegend/397225), CD34 APC (BD/343510), CD14 V450 (BD/561390), and CD206 BV711 (Biolegend/321136); to exclude the T-cells, CD3 BV421 (BD/562427) and CD8 BV421 (Biolegend, 344748) were used. Conventional flow cytometry was performed on a BD FACS LSR-II 4L Full (BD Biosciences, San Jose, CA, USA) using BD FACSDiva version 6 software. Spectral flow cytometry was performed on a 5L Cytek Aurora flow cytometer (Cytek Biosciences, Fremont, CA, USA). Raw spectral flow cytometry data were unmixed using SpectroFlo (Cytek Biosciences). Data were analyzed using OMIQ flow cytometry software (Dotmatics, www.omiq.ai).

### 2.7. Isolation of Neopeptide-Specific CD8 T-Cells

The PBMCs were stained with a mix of PE-conjugated pHLA tetramers for 1 h at 4 °C, washed, and then isolated using magnetic anti-PE beads according to the manufacturers’ instructions (Miltenyi Biotec). Isolated T-cells were stained with antibodies, and single pHLA-tetramer-positive CD8 T-cell clones were sorted with a BD FACSAria III cell sorter (BD Biosciences) using BD FACSDiva version 6 software in a 96-well round-bottomed plate containing 5 × 10^4^ irradiated PBMCs (40 Gy) in 100 µL TCM with 0.4 µg/mL PHA.

### 2.8. T-Cell Reactivity Assays

T-cell clones (4000 T-cells/well) were tested for their reactivity against various stimulator cells (10,000 cells/well) after overnight co-incubation assays in 384-well plates via an IFNγ ELISA (R&D Systems, Minneapolis, MN, USA). For measuring reactivity against AML cell lines and patient samples with *DNMT3A^R882H^* or *IDH2^R140Q^* mutations, higher T-cell numbers (50,000 T-cells/well) were co-incubated with stimulator cells (100,000 cells/well) in 384-well plates, and the IFNγ release was measured in another ELISA (Diaclone, Besançon, France). The AML samples were thawed one day before the flow cytometry and co-incubation with T-cells.

In flow-cytometry-based killing assays, the AML samples were co-incubated with T-cells directly after thawing in transparent IMDM with FBS, human serum, L-glutamine, pen/strep, and 100 IU/mL IL-2. Target cells (50,000 cells/well) were co-incubated with T-cells (50,000 or 150,000 cells/well) in duplicate for 48 h in 96-well plates. Cells were washed and stained with Zombie-Red viability dye (Biolegend, San Diego, CA, USA) for 15 min at room temperature followed by washing and incubation in 10 µL PBS with 2.5% human serum (block). After 15 min, the cells were stained with antibodies for 30 min at 4 °C, washed, and resuspended in 85 μL. A 30 µL amount of each sample was recorded on a 5L Cytek Aurora flow cytometer (Cytek Biosciences, Fremont, CA, USA). Specific lyses were determined by measuring the number of viable target cells per sample. T-cells were excluded from this analysis.

### 2.9. Statistical Analyses

Statistical analyses were performed using Prism software V9.3.1. (GraphPad Software, San Diego, CA). *p*-values < 0.05 were considered significant. The mean numbers of alive cells after the co-incubation killing assays were compared with unpaired *t*-tests adjusted for multiple comparison according to the Benjamini method. Significance levels are indicated as * *p* < 0.05, ** *p* < 0.01, *** *p* < 0.001, and **** *p* < 0.0001.

## 3. Results

To identify HLA class I-binding neoantigens encoded by *DNMT3A* and *IDH1/2* hotspot mutations in AML, we followed a multi-step screening approach, as outlined in Figure 1.

### 3.1. Identification of DNMT3A^R882H^ and IDH2^R140Q^ Neopeptides Using HLA Class I Peptidomics

To investigate whether *DNMT3A*, *IDH1*, or *IDH2* hotspot mutations encode neopeptides that can be processed and presented on the cell surface, a retroviral construct containing these mutations as minigenes was introduced in EBV-B cell lines expressing common HLA class I alleles [21] (Appendix A), and peptides presented by HLA class I on the cell surface were analyzed using data-dependent acquisition tandem mass spectrometry (DDA-MS). Two peptides derived from DNMT3A^R882H^ (11mer YTDVSNMSHLA and 9mer YTDVSNMSH) and one peptide derived from IDH2^R140Q^ (10mer SPNGTIQNIL) were eluted and identified via MS/MS (Figure 2A,B and Appendix A). No peptides derived from DNMT3A^R882C^, IDH1^R132C^, IDH1^R132G^, IDH1^R132S^, IDH1^R132H^, or IDH2^R172K^ were identified.

YTDVSNMSHLA, YTDVSNMSH, and SPNGTIQNIL were validated by comparing the mass spectra of the eluted peptides with those of the synthetic peptides. The peptides also showed predicted HLA class I-binding when analyzed using NetMHCpan 4.1 (Appendix A). DNMT3A^R882H^-derived YTDVSNMSHLA and YTDVSNMSH were predicted to bind to HLA-A*01:01 with weak (0.701% Rank) and strong (0.107% Rank) affinity, respectively. IDH2^R140Q^-derived SPNGTIQNIL was predicted as strong binder for HLA-B*07:02 (0.167% Rank). The wildtype IDH2 peptide SPNGTIRNIL (0.027% Rank) showed stronger predicted HLA-B*07:02 binding than the IDH2^R140Q^ peptide, whereas the predicted HLA-A*01:01 binding of the wildtype DNMT3A peptides (YTDVSNMSRLA: 1.089% Rank; YTDVSNMSR: 0.438% Rank) was lower than that of the DNMT3A^R882H^ peptides. An additional 43 peptides encoded by *DNMT3A*, *IDH1*, and *IDH2* hotspot mutations were predicted binders but not identified in our peptide elutions.

To investigate whether DNMT3A^R882H^ and IDH2^R140Q^ peptides are presented by HLA class I on AML, we performed parallel reaction-monitoring mass spectrometry (PRM-MS), which is a sensitive method for detecting an eluted peptide relative to a spiked-in, heavy-isotope labeled synthetic reference peptide. PRM-MS was performed on HLA class I peptide eluates from the AML cell lines and patient-derived AML samples. SET-2 is an AML cell line with a natural heterozygous *DNMT3A^R882H^* mutation [16], while K562-R140Q is a chronic myelogenous leukemia K562 derivative with homozygous *IDH2^R140Q^* mutations introduced by CRISPR-Cas9-mediated genome editing [17]. Sanger sequencing confirmed the presence of *DNMT3A^R882H^* and *IDH2^R140Q^* mutations in these cell lines (Appendix A). In our PRM-MS analysis, we could not detect the 11mer YTDVSNMSHLA peptide on SET-2 transduced with HLA-A*01:01, nor on HLA-A*01:01-positive patient-derived AML cells with a *DNMT3A^R882H^* mutation (AML 8278) (Appendix A). However, YTDVSNMSHLA was detected on SET-2 transduced with HLA-A*01:01 and the full-length *DNMT3A^R882H^* gene (Figure 2C). In contrast to YTDVSNMSHLA, the 9mer YTDVSNMSH was not detected in any of the HLA eluates (Appendix A).

Similarly, using PRM-MS, the IDH2^R140Q^ -derived SPNGTIQNIL peptide was detected on K562-R140Q cells transduced with HLA-B*07:02 and the full-length *IDH2^R140Q^* gene (Figure 2D), but not on K562-R140Q cells transduced with HLA-B*07:02, nor on HLA-B*07:02-positive patient-derived AML cells with an *IDH2^R140Q^* mutation (AML 9448) (Appendix A). We also performed DDA-MS on patient-derived AML cells with an *IDH2^R140Q^* mutation and detected the wildtype IDH2-derived SPNGTIRNIL peptide, but not IDH2^R140Q^ SPNGTIQNIL. There was no significant difference in the HLA class I surface expression between SET-2 and K562-R140Q and the two patient-derived AML samples (Appendix A). High variant allele frequencies of *DNMT3A^R882H^* and *IDH2^R140Q^* were detected in the patient-derived AML samples via RNA-Seq, and the mutations were validated using Sanger sequencing [18].

Our data showed that YTDVSNMSHLA and SPNGTIQNIL are neopeptides that can be processed and presented by HLA class I on cell lines overexpressing the mutated oncoproteins after retroviral transduction. Surface presentation of the neopeptides on the AML cell lines and patient-derived AML samples carrying endogenous *DNMT3A^R882H^* or *IDH2^R140Q^* mutations, however, could not be detected with HLA class I immunopeptidomics. Since full-length mutant genes were introduced into the same cell lines that expressed the mutations endogenously, this difference in the surface presentation of neopeptides between the gene-transduced and parental cell lines as detected with mass spectrometry is most likely caused by the levels of antigen expression.

### 3.2. Isolation of DNMT3A^R882H^ and IDH2^R140Q^ Neopeptide-Specific T-Cells

Since mass spectrometry showed that YTDVSNMSHLA and SPNGTIQNIL are neopeptides that can be processed and presented on the cell surface, we investigated whether T-cells can react against AML with *DNMT3A^R882H^* or *IDH2^R140Q^* mutations, as T-cells allow the recognition of only a few pHLA complexes.

A total of 26 individuals were screened for DNMT3A^R882H^-specific T-cells (Appendix A), and 30 individuals were screened for IDH2^R140Q^-specific T-cells (Appendix A). PBMCs (0.3–1.7 billion cells) were incubated with PE-conjugated pHLA tetramers (DNMT3A-YTD11-A1 and/or IDH2-SPN-B7), and tetramer-positive CD8 T-cells were enriched with magnetic anti-PE beads and single-cell-sorted using flow cytometry (Appendix A). Growing T-cell clones were tested against EBV-B cells pulsed with YTDVSNMSHLA or SPNGTIQNIL via an IFN-γ ELISA, and peptide-specific T-cell clones were subsequently tested for their reactivity against K562 and HL-60 cells transduced with full-length mutant or wildtype *DNMT3A* or *IDH2* genes. For DNMT3A, 43 T-cell clones were isolated that were peptide-specific (Appendix A). Five of these T-cell clones reacted against K562 transduced with HLA-A*01:01 and the *DNMT3A^R882H^* gene. These clones were also reactive against HL-60, which is an HLA-A*01:01-positive AML cell line, transduced with the *DNMT3A^R882H^* gene. K562 and HL-60 transduced with the wildtype *DNMT3A* gene were not recognized (Figure 3).

For IDH2, five T-cell clones were isolated that were peptide-specific, and three clones reacted against K562 transduced with HLA-B*07:02 and the *IDH2^R140Q^* gene, but not against K562 transduced with the wildtype *IDH2* gene (Figure 4).

### 3.3. T-Cell Reactivity against AML with DNMT3A^R882H^ or IDH2^R140Q^ Mutations

Next, we tested whether the isolated T-cell clones were able to recognize AML cells with *DNMT3A^R882H^* or *IDH2^R140Q^* mutations. All isolated T-cell clones for DNMT3A^R882H^ were reactive against SET-2 transduced with HLA-A*01:01 and the *DNMT3A^R882H^* gene, while only one T-cell clone (8.6F10) showed reactivity against SET-2 transduced with HLA-A*01:01 (Figure 5A). This clone was also reactive against patient-derived AML cells (AML 8278) naturally expressing *DNMT3A^R882H^*, whereas it failed to react against HLA-A*01:01-positive patient-derived AML cells with wildtype *DNMT3A* (AML 5205). All SET-2 cell lines and patient samples expressing HLA-A*01:01 were strongly recognized by an alloreactive HLA-A*01:01-specific T-cell clone. We extended the panel of patient samples and demonstrated that clone 8.6F10 reacted against all HLA-A*01:01 and *DNMT3A^R882H^*-positive AML samples, whereas it again failed to react against AML cells with wildtype *DNMT3A* (Figure 5B and Appendix A).

For IDH2^R140Q^-specific T-cell clones, no reactivity was observed against K562-R140Q transduced with HLA-B*07:02 or patient-derived AML cells (AML 9448) with *IDH2^R140Q^*, while all T-cell clones reacted against K562-R140Q cells transduced with HLA-B*07:02 and the full-length *IDH2^R140Q^* gene (Figure 5C). All patient-derived AML samples expressed HLA class I and costimulatory molecules (Appendix A) and were strongly recognized by an alloreactive HLA-B*07:02-specific T-cell clone (Figure 5C).

### 3.4. DNMT3A^R882H^ Encodes a Neoantigen That Can Be Targeted in HLA-A*01:01-Positive AML

As T-cell clone 8.6F10 showed a specific recognition of AML cells in the IFN-γ ELISA, we next investigated its capacity to target AML in flow-cytometry-based cytotoxicity experiments. After 48 h of co-incubation, T-cell clone 8.6F10 induced a specific lysis of SET-2 transduced with HLA-A*01:01 and the *DNMT3A^R882H^* gene (Figure 6A). Upon the introduction of HLA-A*01:01, parental SET-2 cells naturally expressing *DNMT3A^R882H^* were also killed. HLA-A*01:01-transduced SET-2 cells were also lysed by an alloreactive HLA-A*01:01-specific T-cell clone. SET-2 cells lacking HLA-A*01:01 were not killed, and no lysis was observed upon co-incubation with an HLA-A*01:01-restricted control T-cell clone for CMVpp50.

T-cell clone 8.6F10 was also tested for its capacity to target patient-derived AML cells. Using hematopoietic stem cell (CD34) and myeloid maturation (CD14 and CD206) markers, three HLA-A*01:01-positive patient-derived AML samples with *DNMT3A^R882H^* were shown to contain mixes of immature and mature malignant cell populations (Figure 6B). AML 8278 contained immature CD34-positive and more mature CD34-negative cell populations, whereas AML 16072 consisted of a mix of immature and mature cell populations with low and high CD14 expressions, respectively. AML 14235 had cell populations with different expressions of CD206, which is a known maturation marker for M2 macrophages and dendritic cells [22,23]. After 48 h of co-incubation, T-cell clone 8.6F10 showed specific lysis of all three patient samples, with mature AML cells being killed more strongly than immature AML cells (Figure 6C). Patient-derived AML cells were also killed by the HLA-A*01:01-specific alloreactive T-cell clone, whereas no lysis was observed upon co-incubation with a CMV-specific control T-cell clone.

In conclusion, we isolated a T-cell clone for YTDVSNMSHLA showing a specific recognition and lysis of patient-derived AML cells, thereby validating this peptide as a neoantigen that can be targeted in HLA-A*01:01-positive AML with *DNMT3A^R882H^*.

## 4. Discussion

AML cells often harbor hotspot mutations in the *DNMT3A*, *IDH1*, or *IDH2* genes involved in DNA (hydroxy)methylation. In this study, we explored whether these AML hotspot mutations encode HLA class I-binding neoantigens. Immunopeptidomics via tandem mass spectrometry revealed three peptides that can be processed and presented on the cell surface. For two of these peptides, i.e., DNMT3A^R882H^-derived YTDVSNMSHLA and IDH2^R140Q^-derived SPNGTIQNIL, pHLA tetramers were generated to search for specific T-cells. Of all isolated clones, one T-cell clone for DNMT3A^R882H^ showed specific recognition and lysis of patient-derived AML samples, thereby confirming the peptide as a neoantigen in AML that has the potential to be targeted with immunotherapy. 

Of the various single-amino-acid substitutions in DNMT3A-R882, IDH1-R132, IDH2-R140, and IDH2-R172, 46 peptides showed predicted binding to common HLA class I alleles when analyzed using NetMHCpan4.1. Three of these peptides were identified with DDA-MS on the EBV-B cell lines in which hotspot mutations were introduced as minigenes. Using PRM-MS, which is a highly sensitive technique that allows the detection of a few pHLA surface complexes [24], two of these peptides were also shown to be present on AML cell lines transduced with the mutant gene. However, the surface presentation of these neopeptides could not be directly demonstrated on patient-derived AML cells with the mutation. Other studies have also used mass spectrometry and demonstrated that IDH2^R140Q^-derived SPNGTIQNIL is present on COS-7 cells transfected with HLA-B*07:02 and the *IDH2^R140Q^* gene [25]. In contrast to SPNGTIQNIL, we were able to detect the wildtype IDH2 peptide SPNGTIRNIL on patient-derived AML cells, which is in line with the stronger HLA-B*07:02 binding predicted for this peptide (Appendix A). Despite it not being detected with mass spectrometry, we successfully isolated a T-cell clone that specifically reacts to HLA-A*01:01- and *DNMT3A^R882H^*-positive patient-derived AML samples, thereby confirming that YTDVSNMSHLA is an HLA-A*01:01-binding neoantigen. This demonstrates that although HLA class I peptidomics via mass spectrometry is a reliable technique for measuring neopeptides, a lack of detection does not necessarily indicate absence on the cell surface.

To investigate whether neoantigens derived from DNMT3A^R882H^ or IDH2^R140Q^ can be recognized on target cells by T-cells, we isolated and tested T-cell clones sequentially against cell types increasingly resembling patient-derived AML cells. The clones were tested stepwise against EBV-B cells pulsed with mutant or wildtype peptides, cell lines transduced with full-length mutant or wildtype genes, AML cell lines carrying the mutation naturally or with the mutation introduced by CRISPR-Cas9, and finally against patient-derived AML samples with the mutation. Of the 43 and 5 T-cell clones reacting against EBV-B cells pulsed with exogenous YTDVSNMSHLA or SPNGTIQNIL peptides, 5 and 3 T-cell clones recognized cell lines transduced with the *DNMT3A^R882H^* or *IDH2^R140Q^* gene, respectively. In a recent study by Leung et al. [10], PBMCs from 17 healthy donors were screened for T-cells recognizing neopeptides derived from 14 hotspot mutations in AML. The T-cells were stimulated with mature monocyte-derived dendritic cells pulsed with mixes of 15-mer overlapping peptides. T-cells were detected against IDH1^R132H^ peptides in 7 donors, against FLT3^D835Y^ peptides in 6 donors, and against IDH2^R140Q^ peptides in 11 donors, including SPNGTIQNIL. T-cell recognition of patient-derived AML cells has been demonstrated for two IDH2^R140Q^ peptides, i.e., IQNILGGTVF in HLA-B*35:43 and TIQNILGGTV in HLA-B*15:01, but not for SPNGTIQNIL in HLA-B*07:02. T-cell recognition of cell lines overexpressing the *IDH2^R140Q^* gene has also been shown by Hwang et al. [26], who developed CAR-T-cells containing a TCR mimic antibody domain specifically targeting SPNGTIQNIL in HLA-B*07:02. These CAR-T-cells exhibited reactivity against the exogenous peptide with low avidity (EC50 ~10 μg/mL). In comparison, our T-cell clones, particularly clone 1.3H12, displayed a higher avidity (EC50 44 ng/mL). Despite this higher avidity, our three T-cell clones failed to react against *IDH2^R140Q^* patient-derived AML cells, as well as K562 cells engineered with a homozygous *IDH2^R140Q^* mutation through CRISPR-Cas9 genome editing. Our data indicate that T-cell clones capable of recognizing AML cell lines overexpressing the mutant gene often fail to react against patient-derived AML cells. This underscores the importance of validating neoantigens by demonstrating T-cell recognition of tumor cells carrying an endogenous mutation.

Though efficient, our strategy to identify HLA class I-restricted neoantigens encoded by recurrent mutations in AML also has some limitations. First, we searched for specific T-cell clones using pHLA tetramers that were produced for neopeptides identified with mass spectrometry on five selected EBV-B cell lines transduced with a minigene construct with *DNMT3A*, *IDH1*, and *IDH2* hotspot mutations. Three HLA class I-binding neopeptides were successfully identified, but other neopeptides may have been missed due to inaccurate expressions of the hotspot mutation, inefficient processing and presentation of the neopeptide, or an absence of the relevant HLA allele on the selected EBV-B cell lines. Second, by screening thirty healthy donors, three T-cell clones were isolated that recognized K562 transduced with HLA-B*07:02 and the *IDH2^R140Q^* gene. These T-cell clones failed to react against patient-derived AML samples with the relevant HLA and mutation. However, based on these observations, SPNGTIQNIL cannot be excluded as an IDH2^R140Q^ neoantigen in AML, since the affinity of our isolated T-cell clones may have been too low to detect the surface expression of this neoantigen. Similarly, based on the finding that T-cell clones for DNMT3A^R882H^ did not efficiently kill all patient-derived AML cells in vitro, it cannot be concluded that the surface expression of YTDVSNMSHLA on AML cells is too low to be effectively targeted with immunotherapy.

Of the five T-cell clones with specific reactivity against cell lines transduced with *DNMT3A^R882H^*, one T-cell clone was able to recognize and lyse HLA-A*01:01-positive *DNMT3A^R882H^* patient-derived AML cells, demonstrating that YTDVSNMSHLA is a neoantigen in AML. *DNMT3A^R882H^* is considered a founder mutation that is often present in leukemic stem cells, as well as in pre-leukemic hematopoietic stem cells in clonal hematopoiesis [13]. Therefore, immunotherapy targeting YTDVSNMSHLA in AML may be an effective treatment to prevent relapse. T-cell clone 8.6F10 showed high peptide avidity (EC50 1.33 ng/mL) and specific recognition and lysis of patient-derived AML cells, especially of mature subpopulations. Despite this reactivity profile, we consider the affinity of the TCR to be too low for clinical development, given that clone 8.6F10 did not efficiently kill AML subpopulations with more immature phenotypes. Nevertheless, our data confirm the surface presentation of YTDVSNMSHLA on *DNMT3A^R882H^* AML cells, underscoring the potential significance of this neoantigen as a target for immunotherapy, for instance by treating HLA-A*01:01-positive patients with *DNMT3A^R882H^* AML with high-affinity TCRs or TCR mimic antibodies incorporated into CAR-T-cells or employed in bispecific T-cell engagers [27,28].

## Figures and Tables

**Figure 1 biomedicines-12-01086-f001:**
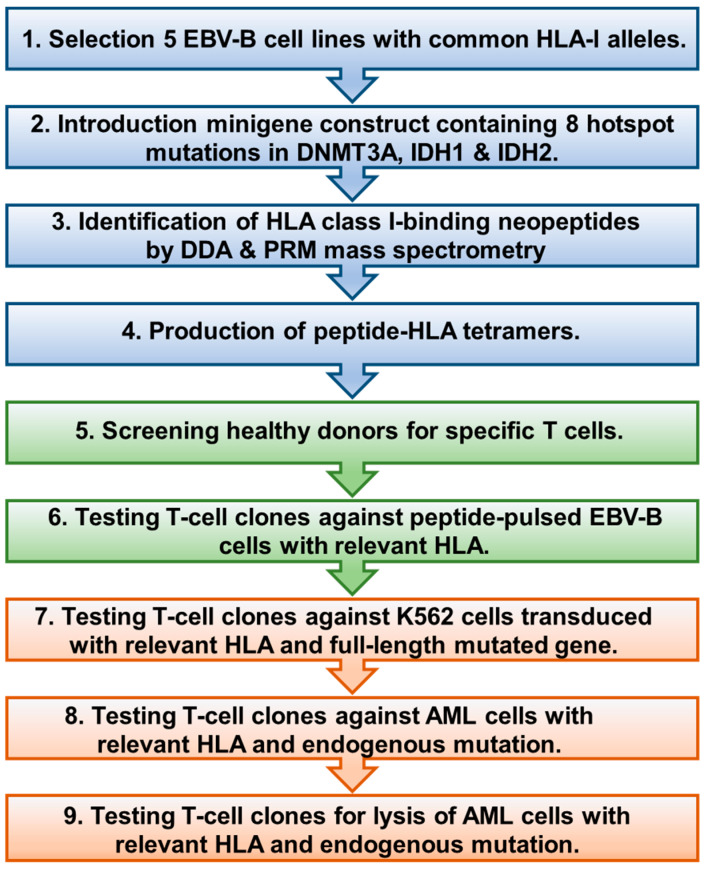
Workflow for identifying DNMT3A, IDH1, or IDH2 neoantigens in AML.

**Figure 2 biomedicines-12-01086-f002:**
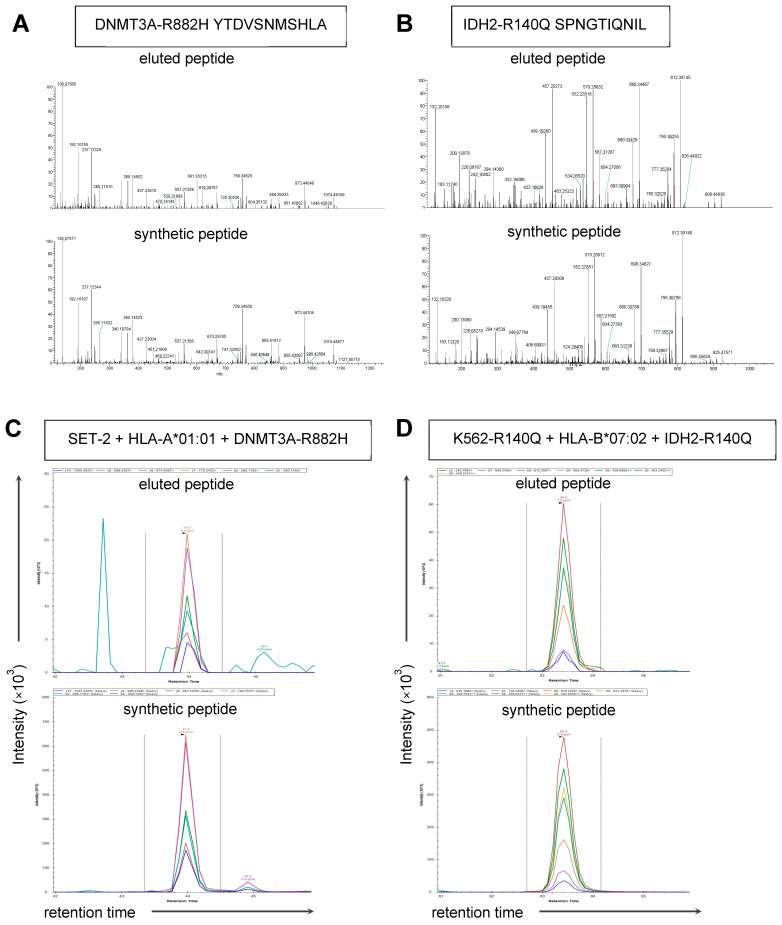
Surface presentation of DNMT3A and IDH2 neopeptides assessed with HLA class I immunopeptidomics. Five EBV-B cell lines expressing common HLA class I alleles transduced with a minigene construct were used to search for DNMT3A and IDH2 neopeptides via HLA class I peptidomics. (**A**) The HLA-A*01:01-binding DNMT3A^R882H^-derived YTDVSNMSHLA peptide was eluted from transduced EBV-B cell line HG00265. The peptide sequence was identified with MS/MS and validated by comparing the eluted (upper panel) and synthetic (lower panel) peptide mass spectra. (**B**) The HLA-B*07:02-binding IDH2^R140Q^-derived SPNGTIQNIL peptide was eluted from transduced EBV-B cell line HG00337. MS/MS spectra are shown for the eluted (upper panel) and synthetic (lower panel) peptide. (**C**) Mass chromatogram for the HLA-A*01:01-binding DNMT3A^R882H^ YTDVSNMSHLA peptide according to PRM-MS. An HLA eluate from the AML cell line SET-2 transduced with HLA-A*01:01 and the full-length *DNMT3A^R882H^* gene was injected into the mass spectrometer. YTDVSNMSHLA (light) was detected at 44 min (top panel). A 20 fmol labeled amount of YTDVSNMSH**L**A (heavy amino acid shown in bold and underlined) was spiked in as a reference and showed the same retention time and the same relative fragment ion intensities (bottom panel). The colored lines represent the relative abundance of the transitions from the precursor ion to the indicated fragment ion. Fragment identities are shown in the boxes. (**D**) Mass chromatogram of the HLA-B*07:02-binding IDH2^R140Q^ SPGNTIQNIL peptide according to PRM-MS analysis. An HLA eluate from the cell line K562-R140Q transduced with HLA-B*07:02 and the full-length *IDH2^R140Q^* gene was injected into the mass spectrometer. SPGNTIQNIL (light) was detected at 93 min (top panel). A 20 fmol amount of S**P**GNTIQNIL (heavy amino acid shown in bold and underlined) was spiked in as a reference and showed the same retention time and the same relative fragment ion intensities (bottom panel).

**Figure 3 biomedicines-12-01086-f003:**
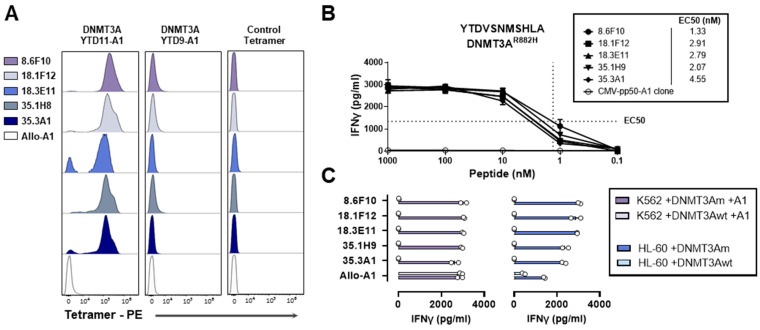
Isolated T-cell clones for HLA-A*01:01-binding DNMT3A^R882H^-derived YTDVSNMSHLA. (**A**) T-cell clones were stained with PE-conjugated pHLA tetramers and measured using flow cytometry. T-cell clones showed specific binding to the HLA-A*01:01 tetramer containing 11mer YTDVSNMSHLA (DNMT3A-YTD11-A1), but not to HLA-A*01:01 tetramers containing 9mer YTDVSNMSH (DNMT3A-YTD9-A1) or an HLA-A*01:01 tetramer containing a CMVpp50 peptide (control tetramer). (**B**) T-cell clones were tested against HLA-A*01:01-positive EBV-B cells exogenously loaded with titrated concentrations of DNMT3A^R882H^-derived YTDVSNMSHLA and measured with an IFN-γ ELISA in duplicate wells after overnight co-incubation. T-cell clone avidity is indicated by the half-maximum effective concentration (EC50), shown with dotted lines for clone 8.6F10 and for all clones in nM. (**C**) T-cell clones were tested against HLA-A*01:01-transduced K562 cells and HLA-A*01:01-positive HL-60 cells transduced with full-length mutant or wildtype *DNMT3A* genes and measured with an IFN-γ ELISA in duplicate wells after overnight co-incubation. All T-cell clones were shown to react against K562 transduced with HLA-A*01:01 and the *DNMT3A^R882H^* gene in two independent experiments.

**Figure 4 biomedicines-12-01086-f004:**
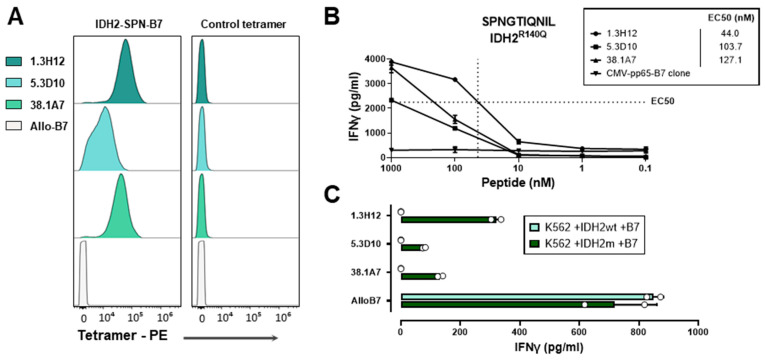
Isolated T-cell clones for HLA-B*07:02-binding IDH2^R140Q^-derived SPNGTIQNIL. (**A**) Specific T-cell clones were stained with PE-conjugated pHLA tetramers and measured using flow cytometry. T-cell clones showed specific binding to the HLA-B*07:02 tetramer containing 10mer SPNGTIQNIL (IDH2-SPN-B7), but not to an HLA-B*07:02 tetramer containing a CMVpp65 peptide (control tetramer). (**B**) T-cell clones were tested against HLA-B*07:02-positive EBV-B cells exogenously loaded with titrated concentrations of IDH2^R140Q^-derived SPNGTIQNIL and measured with an IFN-γ ELISA in duplicate wells after overnight co-incubation. T-cell clone avidity is indicated by the half-maximum effective concentration (EC50), shown with dotted lines for clone 1.3H12 and for all clones in nM. (**C**) T-cell clones were tested against K562 transduced with HLA-B*07:02 and full-length mutant or wildtype *IDH2* genes and measured with an IFN-γ ELISA in duplicate wells after overnight co-incubation. All T-cell clones were shown to react against K562 transduced with HLA-B*07:02 and the mutant *IDH2^R140Q^* gene in two independent experiments.

**Figure 5 biomedicines-12-01086-f005:**
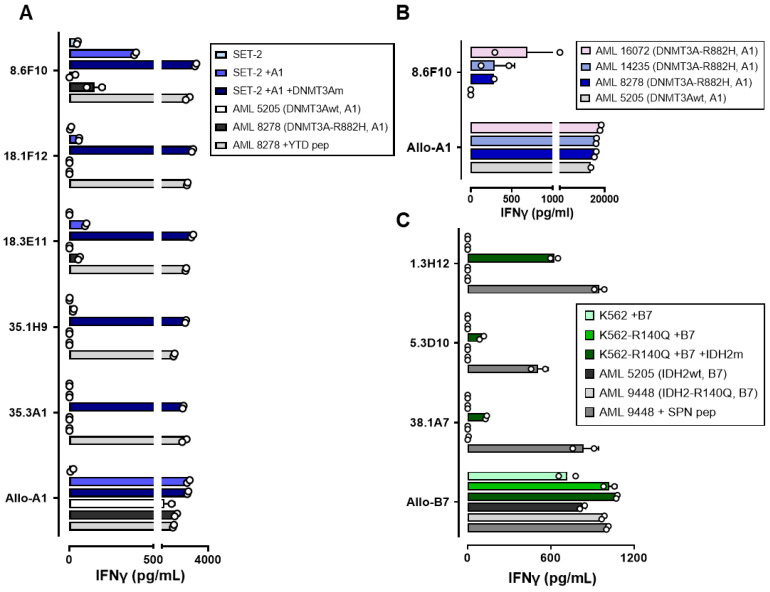
T-cell reactivity against AML with *DNMT3A^R882H^* or *IDH2^R140Q^* mutations. T-cell clones were tested for reactivity against AML cell lines with endogenous *DNMT3A^R882H^* (SET-2) or *IDH2^R140Q^* (K562-R140Q). T-cell reactivity was measured with an IFN-γ ELISA in duplicate wells after overnight co-incubation. (**A**) The reactivity of DNMT3A^R882H^-specific T-cell clones was tested against SET-2 (*DNMT3A^R882H^*), SET-2 transduced with HLA-A*01:01, SET-2 transduced with HLA-A*01:01 and the *DNMT3A^R882H^* gene, and two patient-derived AML samples. AML 8278 was positive for HLA-A*01:01 and *DNMT3A^R882H^*. AML 5205 was HLA-A*01:01-positive, but wildtype *DNMT3A*. T-cell reactivity against AML 8278 was also tested after pulsing with YTDVSNMSHLA, and an alloreactive HLA-A*01:01-specific T-cell clone was included as a control (Allo-A1). (**B**) The reactivity of T-cell clone 8.6F10 was tested against three HLA-A*01:01 and *DNMT3A^R882H^* patient-derived AML samples: 16,072; 14,235; and 8278. AML 5205 (HLA-A*01:01, wildtype *DNMT3A*) was included as a negative control. The results of an independent experiment are shown in Appendix A. (**C**) The reactivity of IDH2^R140Q^-specific T-cell clones was tested against K562 transduced with HLA-B*07:02, K562-R140Q transduced with HLA-B*07:02, K562-R140Q transduced with HLA-B*07:02 and the *IDH2^R140Q^* gene, and two patient-derived AML samples. AML 9448 was positive for HLA-B*07:02 and *IDH2^R140Q^*. AML 5205 was positive for HLA-B*07:02, but wildtype *IDH2*. T-cell reactivity against AML 9448 was also tested after pulsing with SPNGTIQNIL, and an alloreactive HLA-B*07:02-specific T-cell clone was included as a control (Allo-B7).

**Figure 6 biomedicines-12-01086-f006:**
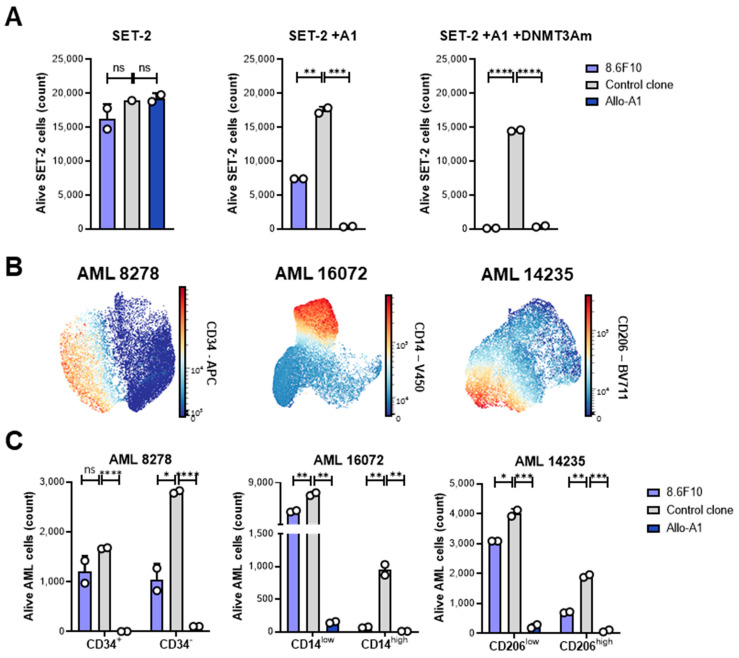
*DNMT3A^R882H^* encodes a neoantigen that can be targeted in HLA-A*01:01-positive AML. (**A**) Number of viable SET-2 cells after 48 h of co-incubation with DNMT3A^R882H^-specific T-cell clone 8.6F10, alloreactive HLA-A*01:01-specific T-cell clone (Allo-A1), or a control T-cell clone specific for the CMVpp50 VTEHDTTLY (VTE) epitope in HLA-A*01:01, as measured with flow cytometry. Viable SET-2 cells negative for Zombie-Red viability staining were gated on expression of CD33 and CD34 and a lack of CD3 and CD8. Killing at effector-to-target ratios of 1:1 was measured in duplicate wells. (**B**) Phenotype analysis of patient-derived AML samples 8278, 16,072, and 14,235 using antibodies against CD34, CD14, and CD206, respectively, as measured with flow cytometry. Viable AML cells negative for Zombie-Red viability staining were gated on expression of CD33. ns = not significant; ** *p* < 0.01; *** *p* < 0.001; **** *p* < 0.0001. (**C**) Number of viable patient-derived AML cells after 48 h of co-incubation with DNMT3A^R882H^-specific T-cell clone 8.6F10, alloreactive HLA-A*01:01-specific T-cell clone (Allo-A1), or CMV-specific control T-cell clone (VTE), as measured with flow cytometry. Specific lyses are separately shown for mature (CD34^−^, CD14^high^, CD206^high^) and immature (CD34^+^, CD14^low^, CD206^low^) AML subsets. Viable AML cells negative for Zombie-Red viability staining were gated on expression of CD33 and a lack of CD3 and CD8. Killing at effector-to-target ratios of 1:1 was measured in duplicate wells. ns = not significant; * *p* < 0.05; ** *p* < 0.01; *** *p* < 0.001; **** *p* < 0.0001.

## Data Availability

The mass spectrometry proteomics data have been deposited in the ProteomeXchange Consortium via the PRIDE [1] partner repository under the dataset identifier PXD050560.

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
