# Peer review of "Hotspot DNA Methyltransferase 3A (*DNMT3A*) and Isocitrate Dehydrogenase 1 and 2 (*IDH1/2*) Mutations in Acute Myeloid Leukemia and Their Relevance as Targets for Immunotherapy"

_biomedicines, 2024, doi:10.3390/biomedicines12051086_

Round 1
Reviewer 1 Report
Comments and Suggestions for Authors
The manuscript biomedicines-2938605 reports the hotspot mutations on DNMT3A and IDH1/2 that immunotherapy can target. The methodology was described in a good way and the results are interesting. Thus, I recommend the publication after minor revision as follows below:
1) The authors considered statistical analysis, however, the plots in Figures 2C, 3C, 4, and 5 show only duplicate assays. Please, is it correct? Clarify it.
2) It would be valuable to briefly mention any limitations of the study and potential directions for future research. Are there any specific aspects of the hotspot mutations on DNMT3A and IDH1/2 that were not addressed and could be explored in future investigations? Please, include it in the manuscript.
3) Please, clarify in the manuscript to use a tetramer peptide as control.
4) A workflow in the manuscript will be interesting to understand the construction of the study better.
5) What is the 3D arrangement (structure) of the peptides? Is there the possibility to conduct circular dichroism to evaluate it?
Author Response
1.1. The authors considered statistical analysis, however, the plots in Figures 2C, 3C, 4, and 5 show only duplicate assays. Please, is it correct? Clarify it.
Reply to 1.1
In the experiments shown in Figures 3C and 4C of the revised manuscript (Figures 2C and 3C in the original manuscript), T-cell clones from different donors have been thawed, restimulated, and tested simultaneously to compare reactivity in the same experiment. However, during our screening and selection approach, all T-cell clones have been tested multiple times for reactivity against K562 transduced with full-length mutant or wildtype DNMT3A or IDH2 genes in separate experiments. This information has been added to the legends of Figures 3 and 4 in the revised manuscript.
In the experiments shown in Figure 5 of the revised manuscript (Figure 4 in the original manuscript), T-cell clones have been tested against cell lines and patient-derived AML samples carrying endogenous DNMT3AR882H or IDH2R140Q mutations. Although only duplicate assays are shown in Figure 4, similar results have been obtained in independent experiments. For T-cell clone 8.6F10, which is the only clone able to react against patient-derived AML samples, data from the independent experiment have been added as Figure S6 to the revised manuscript.
In the experiments shown in Figure 6 of the revised manuscript (Figure 5 in the original manuscript), lysis of AML cell line SET-2 transduced with HLA-A*01:01 and patient-derived AML samples carrying endogenous DNMT3AR882H mutations by T-cell clone 8.6F10 is shown. Although Figure 5 shows only duplicate assays at effector-to-target ratios of 1:1, similar results have been obtained at effector-to-target ratios of 3:1. However, since a small percentage of non-specific lysis was observed for parental SET-2 cells at effector-to-target ratios of 3:1, we decided to exclude the data from the manuscript.
1.2. It would be valuable to briefly mention any limitations of the study and potential directions for future research. Are there any specific aspects of the hotspot mutations on DNMT3A and IDH1/2 that were not addressed and could be explored in future investigations? Please, include it in the manuscript.
Reply to 1.2
Our strategy to identify HLA class I-restricted neoantigens encoded by recurrent mutations in AML has indeed also some limitations. First, we searched for specific T-cell clones using pHLA tetramers that were produced for neopeptides identified by mass spectrometry on five selected EBV-B cell lines transduced with a minigene construct with DNMT3A, IDH1, and IDH2 hotspot mutations. Three HLA class I-binding neopeptides were successfully identified, but other neopeptides may have been missed due to inaccurate expression of the hotspot mutation, inefficient processing and presentation of the neopeptide, or absence of the relevant HLA allele on the selected EBV-B cell lines. Second, by screening 30 healthy donors, three T-cell clones were isolated recognizing K562 transduced with HLA-B*07:02 and the IDH2R140Q gene. These T-cell clones failed to react against patient-derived AML samples with the relevant HLA and mutation. However, based on these observations, SPNGTIQNIL cannot be excluded as neoantigen on AML, since the affinity of our isolated T-cell clones may have been too low to detect surface expression of the neoantigen. Similarly, based on the finding that T-cell clones for DNMT3AR882H did not efficiently kill all patient-derived AML cells in vitro, it cannot be concluded that surface expression of YTDVSNMSHLA on AML is too low to be effectively targeted by immunotherapy.
On request of the reviewer, we added a paragraph describing the limitations of our study in the Discussion of the revised manuscript (lines 494-510).
1.3. Please, clarify in the manuscript to use a tetramer peptide as control.
Reply to 1.3
As negative controls, T-cell clones for DNMT3AR882H were stained with an HLA-A*01:01 tetramer containing a CMVpp50 peptide. T-cell clones for IDH2R140Q were stained with an HLA-B*07:02 tetramer containing a CMVpp65 peptide. This information has been added to the legends of Figures 3 and 4 in the revised manuscript (Figures 2 and 3 in the original manuscript).
1.4. A workflow in the manuscript will be interesting to understand the construction of the study better.
Reply to 1.4
On request of the reviewer, we added a new Figure (Figure 1 in the revised manuscript) to show our workflow to identify DNMT3A, IDH1, and IDH2 neoantigens on AML.
1.5. What is the 3D arrangement (structure) of the peptides? Is there the possibility to conduct circular dichroism to evaluate it?
Reply to 1.5
The study aimed to identify neoantigens on AML that can be targeted by T-cells and may therefore be relevant for immunotherapy. To our knowledge, evaluating 3D structures of HLA-peptide complexes requires protein purification followed by other techniques such as crystallization or circular dichroism. Though elucidating the 3D structures of our HLA-peptide complexes is interesting and may give relevant insight, we are no experts in this field and consider these experiments beyond the scope of the manuscript.
Reviewer 2 Report
Comments and Suggestions for Authors
I have reviewed manuscript entitled as "Hotspot DNMT3A and IDH1/2 Mutations in Acute Myeloid Leukemia and Their Relevance as Targets for Immunotherapy". Authors must address following suggestions in revised version of manuscript;
1) Avoid use of abbreviations in Abstract section.
2) Expressions like I, we or us must not be used in abstract section.
3) In introduction section, add a paragraph about technologies used in inducing mutations at gene level.
4) Prepare a list of abbreviations used in this manuscript.
5) Authors have operated LUMOS Mass spectrometer but plots of LUMOS and HOMOS are missing?
6) Improve quality of Figure 4.
7) Add limitations of current work.
8) Authors are encouraged to incorporate references from last fives only.
9) Similarity index must be below 19%.
Thank You
Author Response
2.1. Avoid use of abbreviations in Abstract section.
Reply to 2.1
We avoided the use of “AML” and “EBV-B” in the Abstract, and replaced these abbreviations with “acute myeloid leukemia” and “human B-lymphoblastoid”.
2.2. Expressions like I, we or us must not be used in abstract section.
Reply to 2.2
We avoided the use of “we” in the Abstract.
2.3. In introduction section, add a paragraph about technologies used in inducing mutations at gene level.
Reply to 2.3
Upon request of the reviewer, we added a paragraph about CRISPR technologies that can be used to introduce mutations at the gene level in the Introduction (lines 66-78).
2.4. Prepare a list of abbreviations used in this manuscript.
Reply to 2.4
We added a list of abbreviations to our revised manuscript (lines 31-39).
2.5. Authors have operated LUMOS Mass spectrometer but plots of LUMOS and HOMOS are missing?
Reply to 2.5
Although LUMOS and HOMOS plots are powerful tools to understand the electronic properties of molecules and predicting their behavior in various chemical and physical processes, they are not essential for the aim of our research to investigate whether neopeptides are presented by HLA on the cell surface, and are therefore not included in the manuscript.
2.6. Improve quality of Figure 4.
Reply to 2.6
We improved the quality of Figure 4.
2.7. Add limitations of current work.
Reply to 2.7
Upon request of the reviewer, we added a paragraph on the limitations of our current work flow to identify neoantigens on AML in the Discussion (lines 494-510).
2.8. Authors are encouraged to incorporate references from last fives only.
Reply to 2.8
In the revised manuscript, we replaced many references with references published in the last five years.
2.9. Similarity index must be below 19%.
Reply to 2.9
The similarity of our manuscript has been checked prior to peer-review. It aligns with the journal's requirements and has been confirmed with the editorial office.
Reviewer 3 Report
Comments and Suggestions for Authors
"Hotspot DNMT3A and IDH1/2 Mutations in Acute Myeloid Leukemia and Their Relevance as Targets for Immunotherapy" by Struckman et. al., discovers a new antigen peptide YTDVSNMSHLA in AML that can be used further in CAR-T cell therapy in AML patients. The study is well designed and executed. I would recommend accepting the manuscript in its current form for publication. Thank you.
Author Response
We thank the reviewer for the positive comments.
Reviewer 4 Report
Comments and Suggestions for Authors
“Hotspot DNMT3A and IDH1/2 Mutations in Acute Myeloid Leukemia and Their Relevance as Targets for Immunotherapy” by Struckman et al.
Mutations in genes involved in epigenetic regulation, such as DNA methyltransferase 3A (DNMT3A) and isocitrate dehydrogenase 1 and 2 (IDH1/2), are frequently observed in AML. In this study, the authors have investigated whether hotspot mutations in these genes can generate neoantigens that can be targeted by immunotherapy. EBV-B cell lines expressing common HLA class I alleles were transduced with constructs containing hotspot mutations as minigenes and peptides eluted from HLA class I were analyzed by tandem mass spectrometry (MS). Using immunopeptidomics, they identified two neopeptides that can be processed and presented by HLA class I, but only in SET-2 cells (an AML cell line with a natural heterozygous for DNMT3AR882H) transduced with HLA-A*01:01+full-length DNMT3AR882H, and K562-R140Q cells transduced with HLA-B*07:02+full-length IDH2R140Q, i.e. in cells overexpressing the mutated oncoproteins, whereas in AML patient cells, expressing the endogenous form of the mutated oncoproteins, these peptides are not detected by MS. Despite this, the two peptides, YTDVSNMSHLA for DNMT3AR882H and SPNGTIQNIL for IDH2R140Q, were used to screen for specific T-cell clones in PBMCs from healthy individuals. The authors identified several clones with specific T cell reactivity measured by IFN-ϒ ELISA. Interestingly, among them, one T-cell clone 8.6F10 showed specific recognition and lysis of AML patient cells with DNMT3AR882H mutation, thereby validating surface presentation of the neoantigens. Overall, the manuscript presents a well-conducted experimental design and includes the required controls. The results are properly presented and discussed, highlighting the complexity and challenges of peptidomics studies in the search for cancer immunotherapies. I only have a few minor comments to make.
1. Lines 58-59: The meaning of “shared neoantigens” should be explained more in detail by the authors.
2. Supplementary Materials Figure S2: CML acronym meaning.
3. Line 252: Figure S3 should be Figure S3A
4. PRM mass spectrometry analysis shows that the 11mer YTDVSNMSHLA peptide was not detected on SET-2 cells (natural heterozygous DNMTA3-R882H) transduced with HLA-A*01:01, whereas it was detected on SET-2 cells transduced with HLA-A*01:01 and full-length DNMT3A-R882H gene. The same was observed for K562-R140Q cells. Do the authors attribute this solely to levels of antigen expression or are there other contributing factors?
5. Line 255: Figure S1 should be Figure S1B
6. Line 259: Figure S3 should be Figure S3B
7. Supplementary Materials Figure S5: In the title “-specific CD8 T cells” and in the legend “HLA tetramer-positive CD8 T cells”.
8. Figure 4A shows that all T-cell clones react with SET-2 cell line transduced with HLA-A*01:01 and DNMTA3-R882H. This result should be reported in section 3.3.
9. Figure 5: in the legend report statistical p-values (* p<0.01, ** p…) and ns meaning.
10. Supplementary Materials Figure S6: specify LFA-3 and ICAM-1 and 2 acronym meaning.
In various parts of the manuscript “AML” should be replaced by “AML cells or cell line” where appropriate.
Comments on the Quality of English Language
Minor editing of the English language is required.
Author Response
4.1. Lines 58-59: The meaning of “shared neoantigens” should be explained more in detail by the authors.
Reply to 4.1
We changed “shared neoantigens” to “recurrent neoantigens that are shared between multiple patients” in lines 81-82 of the revised manuscript (lines 58-59 in the original manuscript).
4.2. Supplementary Materials Figure S2: CML acronym meaning.
Reply to 4.2
We explained the acronym CML, which means “chronic myelogenous leukemia”, in the legend of Figure S2 and the Results section.
4.3. Line 252: Figure S3 should be Figure S3A
Reply to 4.3
We refer to Figure S3A in line 283 of the revised manuscript (line 252 in the original manuscript).
4.4. PRM mass spectrometry analysis shows that the 11mer YTDVSNMSHLA peptide was not detected on SET-2 cells (natural heterozygous DNMTA3-R882H) transduced with HLA-A*01:01, whereas it was detected on SET-2 cells transduced with HLA-A*01:01 and full-length DNMT3A-R882H gene. The same was observed for K562-R140Q cells. Do the authors attribute this solely to levels of antigen expression or are there other contributing factors?
Reply to 4.4
We thank the reviewer for this relevant question. We on purpose introduced the full-length mutant DNMT3A and IDH2 genes into the same cell lines that expressed the mutations endogenously to compare surface presentation of the neopeptides by PRM mass spectrometry under artificial conditions of overexpression and more physiological conditions of endogenous expression. To emphasize this more clearly in the revised manuscript, we added the following sentences to the Results section: “Since full-length mutant genes were introduced into the same cell lines that expressed the mutations endogenously, the difference in surface presentation of neopeptides between gene-transduced and parental cell lines as detected by mass spectrometry is most likely caused by levels of antigen expression” (lines 307-310).
4.5. Line 255: Figure S1 should be Figure S1B
Reply to 4.5
We refer to Figure S1B in line 286 of the revised manuscript (line 255 in the original manuscript).
4.6. Line 259: Figure S3 should be Figure S3B
Reply to 4.6
We refer to Figure S3B in line 290 of the revised manuscript (line 259 in the original manuscript).
4.7. Supplementary Materials Figure S5: In the title “-specific CD8 T cells” and in the legend “HLA tetramer-positive CD8 T cells”.
Reply to 4.7
We added “CD8” to the title and in the legend of Figure S5.
4.8. Figure 4A shows that all T-cell clones react with SET-2 cell line transduced with HLA-A*01:01 and DNMTA3-R882H. This result should be reported in section 3.3.
Reply to 4.8
We agree with the reviewer and added this information to lines 357-358 in section 3.3 in the Results of the revised manuscript.
4.9. Figure 5: in the legend report statistical p-values (* p<0.01, ** p…) and ns meaning.
Reply to 4.9
We agree with the reviewer and added this information to the legend of Figure 6 (Figure 5 of the original manuscript).
4.10. Supplementary Materials Figure S6: specify LFA-3 and ICAM-1 and 2 acronym meaning.
Reply to 4.10
We explained the meaning of the acronyms in the legend of Figure S7 (Figure S6 in the original manuscript):
LFA-3; Lymphocyte Function-Associated antigen 3
ICAM-1; InterCellular Adhesion Molecule-1
ICAM-2; InterCellular Adhesion Molecule-2
4.11. In various parts of the manuscript “AML” should be replaced by “AML cells or cell line” where appropriate.
Reply to 4.11
We added “AML cell line(s)” or “patient-derived AML cells” throughout the revised manuscript.
Round 2
Reviewer 2 Report
Comments and Suggestions for Authors
The authors have successfully addressed all of my queries.